Acute sleep fragmentation does not alter pro-inflammatory cytokine gene expression in brain or peripheral tissues of leptin-deficient mice

Dumaine Jennifer E. 1
Ashley Noah T. 2 noah.ashley@wku.edu
1 School of Veterinary Medicine, University of Pennsylvania , Philadelphia, PA , USA
2 Department of Biology, Western Kentucky University , Bowling Green, KY , USA
Brancaleone Vincenzo
Electronic publication date: 2018 Feb 19
Publication date: 2018
Volume: 6
Electronic Location ID: e4423
Received 2017 Oct 30; Accepted 2018 Feb 7
Copyright: © 2018 Dumaine and Ashley
Copyright year: 2018
Copyright holder: Dumaine and Ashley
License: This is an open access article distributed under the terms of the Creative Commons Attribution License, which permits unrestricted use, distribution, reproduction and adaptation in any medium and for any purpose provided that it is properly attributed. For attribution, the original author(s), title, publication source (PeerJ) and either DOI or URL of the article must be cited.
License URL: https://creativecommons.org/licenses/by/4.0/

Keywords: TGF-β1, IL-1β, TNF-α, Ob/ob, Corticosterone, IL-1, Inflammation, Leptin, Sleep loss, Sleep fragmentation

Funding: Institutional Development Award grant from the NIGMS 5P20GM103436, R15GM117534 Graduate Studies Research Grant from Western Kentucky University This work was funded by an Institutional Development Award (IDeA) grant from the NIGMS under grant number 5P20GM103436 (to Noah T. Ashley), NIH R15GM117534 (to Noah T. Ashley) and a Graduate Studies Research Grant from Western Kentucky University (to Jennifer E. Dumaine). The funders had no role in study design, data collection and analysis, decision to publish, or preparation of the manuscript.

==============================
Obesity and sleep fragmentation (SF) are often co-occurring pro-inflammatory conditions in patients with obstructive sleep apnea. Leptin is a peptide hormone produced by adipocytes that has anorexigenic effects upon appetite while regulating immunity. The role of leptin in mediating inflammatory responses to SF is incompletely understood. Male C57BL/6j (lean) and ob/ob mice (leptin-deficient mice exhibiting obese phenotype) were subjected to SF or control conditions for 24 h using an automated SF chamber. Trunk blood and tissue samples from the periphery (liver, spleen, fat, and heart) and brain (hypothalamus, prefrontal cortex, and hippocampus) were collected. Quantitative PCR was used to determine relative cytokine gene expression of pro-inflammatory (IL-1β, TNF-α) and anti-inflammatory (TGF-β1) cytokines. Enzyme-linked immunosorbent assay (ELISA) was used to determine serum corticosterone concentration. Ob/ob mice exhibited elevated cytokine gene expression in liver (TNF-α, TGF-β1), heart (TGF-β1), fat (TNF-α), and brain (hippocampus, hypothalamus, prefrontal cortex: IL-1β, TNF-α) compared with wild-type mice. Conversely, leptin deficiency decreased pro-inflammatory cytokine gene expression in heart (IL-1β, TNF-α). SF significantly increased IL-1β and TNF-α gene expression in fat and TGF-β1 expression in spleen relative to controls, but only in wild-type mice. SF increased basal serum corticosterone regardless of genotype. Taken together, these findings suggest that leptin deficiency affects cytokine gene expression differently in the brain compared to peripheral tissues with minimal interaction from acute SF.

Introduction

In conjunction with the worldwide obesity epidemic, the number of individuals suffering from sleep disorders continues to rise (Peppard et al., 2013). Obesity is a chronic low-grade pro-inflammatory condition, with adipose tissue exhibiting both paracrine and endocrine functions. Leptin is an adipokine that regulates metabolic functions, such as insulin sensitivity and glucose tolerance (Fu et al., 2015; O’Donnell et al., 2000). Leptin is secreted by adipose tissue, where it then circulates to the hypothalamus to act as a satiety hormone. Serum leptin levels are directly related to adipose tissue mass, which leads to increased leptin levels associated with the obese (OB) phenotype (Al Maskari & Alnaqdy, 2006). Although obesity increases serum leptin concentration, there is no corresponding loss of appetite, which suggests resistance of individuals to the actions of this hormone (O’Donnell et al., 2000). In addition to having metabolic functions, leptin regulates the sleep-wake cycle, as leptin-deficient mice exhibit more frequent, shorter-duration sleep periods, increased bouts of REM sleep, and decreased recovery to sleep deprivation (Laposky et al., 2006). Although leptin can act as a pro-inflammatory signal (Besedovsky, Lange & Born, 2012), mice that are deficient in leptin (ob/ob) or possess a defective leptin receptor (db/db and fa/fa) are morbidly OB and display elevated pro-inflammatory cytokines in peripheral tissues and brain (Dube, Torto & Kalra, 2008; Fruhbeck et al., 2017; Mirsoian et al., 2014; Sainz et al., 2010). This suggests that excessive adiposity overrides the absence of leptin to promote a low-grade inflammatory environment in these tissues.

Due to alterations in leptin concentrations following sleep fragmentation (SF) in obstructive sleep apnea (OSA), it has been hypothesized that leptin can be used as a biomarker to detect the presence of sleep disorders (Ozturk et al., 2003; Pan & Kastin, 2014). Despite this proposed relationship, previous attempts have yet to successfully correlate leptin and leptin gene receptor polymorphisms with OSA (Lv et al., 2015). Additional experiments in rats, mice, and humans have reported conflicting results as to whether leptin concentrations increase or decrease following SF (Fu et al., 2015; Garcia-Garcia et al., 2014; Ozturk et al., 2003). In lieu of this variation, it can be concluded that leptin concentrations are affected by both SF and obesity, but their interaction is incompletely understood.

While pathologies associated with OSA are due to chronic exposure, an acute model of SF from our laboratory shows that acute SF rapidly induces characteristic changes in pro-inflammatory cytokine gene expression in the periphery and brain (Dumaine & Ashley, 2015). Interleukin-1-beta (IL-1β) and tumor necrosis factor-alpha (TNF-α) are two major pro-inflammatory cytokines that induce a wide variety of inflammatory events (Dinarello, 2010) whereas transforming growth factor-beta-1 (TGF-β1) is generally considered an anti-inflammatory cytokine that suppresses production of pro-inflammatory cytokines and innate immune cells (Sanjabi et al., 2009). Among C57BL/6j mice, 24 h of SF leads to an upregulation of IL-1β expression in adipose tissue, heart, and hypothalamus, as well as increased expression of TGF-β1 in the hippocampus and prefrontal cortex (Dumaine & Ashley, 2015). Although the simultaneous effects of obesity and SF have been studied within the context of neurobehavior, the effects upon peripheral and neuro-inflammatory responses remain poorly understood (He et al., 2015). The ob/ob knockout (KO) mouse strain was chosen for this experiment to assess the simultaneous effects of SF and the absence of leptin responsiveness, as is the case with a portion of the OSA disease population. We predicted that SF would exacerbate peripheral and brain inflammation in leptin-deficient mice compared with wild-type controls, suggesting an additive effect of SF and leptin deficiency.

Materials and Methods

Animals

Male C57BL/6j mice (n = 16) and ob/ob leptin gene KO mice (background, C57BL/6J; n = 18) were housed in a colony room (12 h light/12 h dark, lights on 07:00 h; 21 ± 1 °C) at Western Kentucky University. To obtain the ob/ob KO mice, Jackson Laboratory ob heterozygotes were purchased and bred. After weaning at 21 days of age, pups resided in polypropylene cages with same sex littermates and were provided corncob bedding and food (Rodent RM4 1800 diet (18% protein, 5% fat, 4% fiber), Cincinnati Lab Supply, Cincinnati, OH, USA) and water ad libitum. Offspring lacking the leptin gene (ob/ob KO) were identified phenotypically by screening individuals for weight gain ca. four weeks of age. This study was conducted under approval of the Institutional Animal Care Committee at Western Kentucky University (#15-11) and procedures followed the National Institutes of Health Guide for the Use and Care of Laboratory Animals and international ethical standards.

Sleep fragmentation

Male adult (>8 weeks of age) ob/ob mice and wild-type mice were briefly exposed to isoflurane vapors and weighed to the nearest 0.01 g, and then placed in an automated SF chamber (Model 80391; Lafayette Instrument Company, Lafayette, IN, USA) with a thin layer of corn cob bedding at 09:00 h. Food and water were provided ad libitum. To distinguish among subjects, tails were marked with a different colored pen (Sharpie, Downers Grove, IL, USA). Following a 23 h habituation period, the swipe bar was activated at 08:00 the next day. For this group, the swipe bar interval was set to 20 s based upon the results of previous experiments using telemetric transmitters to measure Electroencephalogram (EEG) and Electromyogram (EMG) for validation of SF by this method (Ramesh, Kaushal & Gozal, 2009). Thus, each mouse receives 1,400 bar sweeps over 24 h. This value exceeds by an order of magnitude the number of baseline arousals that typically occur in ob/ob and wild-type mice (see Table 2, Laposky et al., 2006). For the control group, the bar remained stationary. Thirty-four mice were pseudo-randomly assigned to either control (n = 8), SF (n = 8), OB control (n = 9), or OB SF groups (n = 9). One mouse in the OB SF group was removed during the experiment due to health reasons. Sample sizes were determined based upon previous experiments (Dumaine & Ashley, 2015).

After 24 h, animals were anesthetized using isoflurane vapors (<2 min), weighed, and rapidly decapitated within 3 min of initial handling. Trunk blood was collected, kept on ice for <20 min, and spun at 3,000g for 30 min at 4 °C. Serum was drawn off and stored at −80 °C for ELISA analysis. Liver, spleen, epididymal white adipose tissue (EWAT), and heart samples were collected and placed into Eppendorf tubes containing RNAlater solution (ThermoFisher Scientific, Waltham, MA, USA). The tissue samples were stored at 4 °C until RNA extraction. The hippocampus, hypothalamus, and prefrontal cortex were dissected from the brain and stored in RNAlater at −20 °C until RNA extraction.

RNA extraction

RNA was extracted from brain, liver, spleen, and fat using an RNeasy Mini Kit (Qiagen, Venlo, Netherlands). RNA was extracted from the heart using an RNeasy Fibrous Tissue Mini Kit. RNA concentration was measured using a NanoDrop 2000 Spectrophotometer (ThermoFisher Scientific, Waltham, MA, USA).

Reverse transcription

Total RNA was reverse transcribed into cDNA using a High-Capacity cDNA Reverse Transcription Kit (#4368813; ThermoFisher Scientific, Waltham, MA, USA). The reaction was carried out according to the manufacturer’s instructions. Amplification conditions for the thermocycler were 25 °C for 10 min, 37 °C for 120 min, 85 °C for 5 min (for one cycle), then 4 °C.

Real time-PCR

Real time-PCR was performed using an ABI 7300 RT-PCR system. To determine relative cytokine gene expression, Taqman Gene Expression RT-PCR Master Mix and the following primer/probes (ThermoFisher Scientific, Waltham, MA, USA) were used: IL-1β (Mm00434228_m1), TNF-α (Mm00443260_g1), and TGF-β1 (Mm01178820_m1), or 18s endogenous primer-limited (Mm03928990_g1). Cytokine probes were labeled with the fluorescent marker 5-FAM at the 5′ end and the quencher MGB at the 3′ end. The 18s endogenous control used a VIC-labeled probe. Amplification conditions were 50 °C for 2 min, 95 °C for 10 min, 40 cycles of 95 °C for 15 s, and 60 °C for 1 min. The reaction was carried out in duplicate according to the manufacturer’s instructions, and the relative expression of IL-1β, TNF-α, and TGF-β1 was determined by comparison with the standard curve generated using serial dilutions of pooled cDNA (1:1, 1:10, 1:100, 1:1,000, 1:10,000) and normalization to the control 18s ribosomal RNA levels and calibrator samples.

ELISA

Serum corticosterone concentration was determined using an ELISA Kit (ab108821; Abcam, Cambridge, UK) with 96% recovery for corticosterone (0.3 ng/mL sensitivity). The reagents and standards were prepared according to the manufacturer’s instructions. Cross reactivities for the kit were <30% deoxycorticosterone, <2% aldosterone, <2% progesterone, and <1% cortexolone. Serum samples were diluted 1:200 and plated in duplicate. The intraassay variation was 5.0% CV.

Statistical analyses

All data were expressed as mean ± standard error. Two-way ANOVAs were used to detect differences among groups, with sleep treatment (control (no SF) vs. SF), genotype (wildtype vs. ob/ob), and the interaction between two, as the main factors, and relative gene expression of each specific cytokine as the dependent variable (α = 0.05). Post hoc tests were carried out using Fisher’s LSD test and using a one-way ANOVA to determine significant interactions (α = 0.05). In some cases, logarithmic transformation was necessary to satisfy requirement of homogeneity of variances.

Results

Body mass

Body mass decreased after 24 h of SF (F1,30 = 42.050; p < 0.0001; Fig. 1). Post-hoc tests revealed that sleep treatment affected mass loss, despite the initial difference in body weight. SF and OB SF mice lost significantly more body mass (on a percentage basis) than wild-type control and OB control mice, respectively (Fisher’s LSD; all p < 0.05; Fig. 1). Wild-type control mice lost significantly more body mass (on a percentage basis) than OB controls (Fisher’s LSD, p < 0.05).

Figure 1 Body mass loss in leptin-deficient (OB) and wild-type (lean) mice exposed to SF or control conditions for 24 h.

Data are shown as % change in body mass ± SE for each group. Shared letters indicate no significant difference between groups. The numbers at the base of the column indicate the sample size of each group.

Peripheral response

Liver

No differences among groups in IL-1β gene expression were detected in the liver (log-transformed; interaction: F2,27 = 0.993; p = 0.328; sleep treatment: F1,27 = 0.809; p = 0.377; leptin deficiency: F1,27 = 2.875; p = 0.102; Fig. 2). Liver TNF-α gene expression was elevated in OB mice relative to wild-type controls (log-transformed; F1,25 = 6.750; p = 0.020; Fisher’s LSD; all p < 0.05; Fig. 2A). No other factors affected TNF-α gene expression (log-transformed; interaction: F2,25 = 1.300; p = 0.260; sleep treatment: F1,25 = 2.080; p = 0.160; Fig. 2A). Leptin deficiency also increased TGF-β1 gene expression in liver (log-transformed; F1,28 = 12.350; p = 0.002; Fisher’s LSD; all p < 0.05; Fig. 2) and no other factors were found to affect TGF-β1 gene expression (log-transformed; interaction: F2,28 = 0.646; p = 0.428; sleep treatment: F1,28 = 1.257; p = 0.272).

Figure 2 Effects of fragmented sleep, leptin deficiency (OB), and their interaction upon cytokine gene expression in peripheral tissues.

(A) Liver, (B) spleen, (C) epididymal adipose tissue, (D) heart. Data are shown as mean ± SE for each group. Sample size ranged from N = 6–9, as several samples were undetectable by RT-PCR in some tissue samples. *Denotes p < 0.05 relative to non-SF vehicle control. †Denotes p < 0.05 relative to group(s) indicated by horizontal bar(s). ‡Denotes p < 0.05 relative to all other groups.

Spleen

Interleukin-1-beta gene expression decreased in the spleen due to leptin deficiency (log-transformed; F1,26 = 6.390; p = 0.02), although after conducting post-hoc tests, only ob SF mice exhibited significantly lower levels than wild-type SF controls (Fisher’s LSD; p < 0.05). No other factors affected IL-1β gene expression (interaction: F2,26 = 0.1.233; p = 0.277; sleep treatment: F1,26 = 0.154; p = 0.698; Fig. 2B). There were no significant differences in TNF-α gene expression in the spleen (log-transformed; interaction: F2,28 = 0.292; p = 0.594; sleep treatment: F1,28 = 0.022; p = 0.883; leptin deficiency: F1,28 = 0.071; p = 0.792; Fig. 2B). SF increased TGF-β1 gene expression in the spleen (log-transformed; F1,25 = 6.820; p = 0.020; Fisher’s LSD; all p < 0.05; Fig. 2B). However, no other factors affected splenic TGF-β1 gene expression (log-transformed; interaction: F2,25 = 1.857; p = 0.185; leptin deficiency: F1,25 = 3.278; p = 0.082; Fig. 2B).

EWAT

Obese mice decreased IL-1β gene expression in EWAT compared with wild-type mice (log-transformed; F1,26 = 6.390; p = 0.018; Fisher’s LSD; all p < 0.05; Fig. 2C). The interaction between sleep treatment and genotype was also significant, with wild-type mice exposed to SF exhibiting elevated IL-1β gene expression relative to other groups (log-transformed; F2,26 = 4.630; p = 0.040; Fisher’s LSD; all p < 0.05; Fig. 2C), except relative to the wild-type controls. TNF-α gene expression was altered in epididymal adipose tissue due to the interaction between sleep treatment and genotype (log-transformed; F2,28 = 8.890; p = 0.006; Fisher’s LSD; all p < 0.05; Fig. 2C). Specifically, wild-type mice subjected to SF significantly increased TNF-α gene expression in EWAT compared with wild-type controls (p = 0.002; Fig. 2), while leptin-deficient mice failed to show such an elevation from SF compared with ob/ob controls (p > 0.05; Fig. 2C). Ob/ob controls had significantly elevated TNF-α gene expression compared with wild-type controls (Fisher’s LSD; p < 0.05; Fig. 2C). No other factors affected TNF-α gene expression in EWAT (log-transformed; sleep treatment: F1,28 = 3.963; p = 0.056; leptin deficiency: F1,28 = 0.095; p = 0.760; Fig. 2C). Similarly, none of the factors tested affected TGF-β1 gene expression in EWAT (interaction: F2,28 = 0.011; p = 0.918; sleep treatment: F1,28 = 0.002; p = 0.962; leptin deficiency: F1,28 = 0.037; p = 0.849; Fig. 2C).

Heart

In cardiac muscle, OB mice exhibited decreased IL-1β (log-transformed; F1,29 = 50.720; p < 0.001; Fisher’s LSD; all p < 0.05; Fig. 2D) and TNF-α gene expression (log-transformed; F1,29 = 28.960; p < 0.001; Fisher’s LSD; all p < 0.05; Fig. 2D) compared with wild-type controls. However, no other factors affected IL-1β gene expression (log-transformed; interaction: F2,29 = 2.876; p = 0.101; sleep treatment: F1,29 = 0.003; p = 0.956; Fig. 2D). Sleep treatment also altered TNF-α gene expression in heart (log-transformed; F1,29 = 16.106; p = 0.0004; Fig. 2D), but there was no interaction effect between sleep treatment and leptin deficiency (log-transformed; interaction: F2,29 = 1.660; p = 0.208; Fig. 2D). Leptin deficiency simultaneously resulted in increased cardiac TGF-β1 gene expression (F1,29 = 7.340; p < 0.001; Fisher’s LSD; all p < 0.05; Fig. 3). No other factors affected TGF-β1 gene expression in the heart (interaction: F2,29 = 1.014; p = 0.322; sleep treatment: F1,29 = 3.275; p = 0.081; Fig. 2D).

Figure 3 Effects of fragmented sleep, leptin deficiency (OB), and their interaction upon cytokine gene expression in the brain.

(A) Hippocampus, (B) hypothalamus, and (C) prefrontal cortex. Data are shown as mean ± SE for each group. Sample size ranged from N = 6–9, as several samples were undetectable by RT-PCR in some tissue samples. *Denotes p < 0.05 relative to non-SF vehicle control. †Denotes p < 0.05 relative to group(s) indicated by horizontal bar(s).

Brain response

Hippocampus

Interleukin-1-beta gene expression was elevated in the hippocampus of ob/ob mice compared with wild-type controls (log-transformed; F1,28 = 9.880; p = 0.004; Fisher’s LSD; all p < 0.05; Fig. 3A). Similarly, leptin deficiency increased TNF-α gene expression in the hippocampus relative to wild-type SF and control mice (log-transformed; F1,29 = 24.540; p < 0.0001; Fisher’s LSD; all p < 0.05; Fig. 3A). There were no other significant effects or interactions in IL-1β or TNF-α gene expression (IL-1β: log-transformed; interaction: F2,28 = 0.165; p = 0.688; sleep: F1,28 = 2.218; p = 0.148; TNF-α: log-transformed; interaction: F2,29 = 4.123; p = 0.052; sleep treatment: F1,29 = 0.001; p = 0.980; Fig. 3A). For hippocampal TGF-β1 gene expression, there were no significant effects or interactions among groups (hippocampus: interaction: F2,27 = 1.116; p = 0.300; sleep treatment: F1,27 = 0.074; p = 0.787; leptin deficiency: F1,27 = 0.127; p = 0.724; Fig. 3A).

Figure 4 Serum corticosterone concentrations after 24 h of sleep fragmentation or control conditions in wild-type (lean) and leptin-deficient (OB) mice.

Data are shown as mean (ng/mL) ± SE for each group. Shared letters indicate there is no significant difference between groups. Numbers at the base of the column indicate sample size.

Hypothalamus

Leptin-deficient mice exhibited elevated levels of hypothalamic IL-1β (log-transformed; F1,26 = 11.640; p = 0.002; Fisher’s LSD; all p < 0.05; Fig. 3B) and TNF-α (log-transformed; F1,25 = 29.270; p < 0.0001; Fisher’s LSD; all p < 0.05; Fig. 3B) compared with wild-type mice. No other significant effects or interactions were apparent (hypothalamus IL-1β: log-transformed; interaction: F2,26 = 0.012; p = 0.914; sleep treatment: F1,26 = 0.002; p = 0.962; TNF-α hypothalamus: log-transformed; interaction: F1.25 = 0.655; p = 0.426; sleep treatment: F2,25 = 1.196; p = 0.285). Lastly, hypothalamic TGF-β1 gene expression did not vary significantly among groups (log-transformed; interaction: F2,25 = 0.214; p = 0.647; sleep treatment: F1,25 = 2.294; p = 0.142; leptin deficiency: F1,25 = 0.035; p = 0.854).

Prefrontal cortex

Leptin-deficient mice exhibited elevated levels of IL-1β (log-transformed; leptin deficiency: F1,27 = 6.90; p = 0.010; Fisher’s LSD; all p < 0.05; Fig. 3C) and TNF-α (log-transformed; leptin deficiency; F1,27 = 5.62; p < 0.024; Fisher’s LSD; all p < 0.05; Fig. 3C) expression in prefrontal cortex compared with wild-type controls. No other factors affected pro-inflammatory cytokine gene expression in the hypothalamus, prefrontal cortex, or hippocampus (prefrontal cortex IL-1β: log-transformed; interaction: F2,27 = 0.264; p = 0.612; sleep treatment: F1,27 = 1.432; p = 0.242; TNF-α prefrontal cortex: log-transformed; interaction: F2,27 = 0.118; p = 0.734; sleep treatment: F1,27 = 0.319; p = 0.577; leptin deficiency: F1,27 = 0.225; p = 0.639; hippocampus IL-1β: log-transformed; interaction: F2,28 = 0.165; p = 0.688; sleep: F1,28 = 2.218; p = 0.148; hippocampus TNF-α: log-transformed; interaction: F2,29 = 4.123; p = 0.052; sleep treatment: F1,29 = 0.001; p = 0.980; all Fig. 3). There were no significant differences in TGF-β1 gene expression in any regions of the brain tested (hypothalamus, prefrontal cortex: interaction: F2,28 = 1.276; p = 0.268; sleep treatment: F1,28 = 0.006; p = 0.939; leptin deficiency: F1,28 = 0.066; p = 0.799).

Serum corticosterone

Serum corticosterone concentration varied significantly among groups due to sleep treatment (F1,30 = 41.100; p < 0.0001; Fisher’s LSD; all p < 0.05; Fig. 4). Specifically, acute SF significantly increased baseline serum corticosterone in wild-type and ob/ob mice compared with wild-type or ob/ob controls (all p < 0.03). However, serum corticosterone concentration did not differ significantly between ob/ob and wild-type mice (F1,30 = 0.777; p = 0.385; Fig. 4) or as a result of the interaction between sleep treatment and genotype (F2,30 = 0.031; p = 0.861; Fig. 4).

Discussion

The findings of this study indicate that leptin-deficient mice produce different cytokine gene responses in the periphery versus the brain and that SF had little or no effect upon these differences. Therefore, the hypothesis that SF exacerbates peripheral and brain inflammation in leptin-deficient mice compared with wild-type controls is not supported. More specifically, leptin deficiency increased pro-inflammatory (IL-1β, TNF-α) cytokine gene expression in all areas of brain measured (hippocampus, hypothalamus, prefrontal cortex; Table 1). These results are consistent with a number of studies that have described hypothalamic inflammation in OB mice (either genetically or diet-induced) as well as presence of neuroinflammation in hippocampus and cortex (Aguilar-Valles et al., 2015; Guillemot-Legris & Muccioli, 2017; Lumeng & Saltiel, 2011). In the periphery, results were more complex (Table 1). In liver, TNF-α was elevated in ob mice compared with wild-type mice, but in other tissues, ob mice exhibited decreased expression of IL-1β in spleen and IL-1β and TNF-α in heart compared with wild-type mice (Table 1). Furthermore, there was a general upregulation of TGF-β1 in liver, spleen, and heart in ob mice compared with wild-type controls. The propensity toward an anti-inflammatory response in important metabolic tissue could be protective, as OB individuals would be at the greatest risk for the development of metabolic abnormalities (Youn et al., 2014). The increase in TGF-β1 and decrease in pro-inflammatory cytokine (IL-1β, TNF-α) expression in heart of ob mice was unexpected and potentially indicative of an anti-inflammatory environment that may also be protective, although additional study is required. This result contrasts with previous research that shows that leptin-deficiency increases inflammation and hypertrophy in the mouse heart (Unsold et al., 2015).

Table 1 Summary of the effects of leptin deficiency and sleep fragmentation on cytokine gene expression in peripheral and brain tissues.

Tissue	Effect	Factor	
Liver	Increase: TNF-α and TGF-β1	Leptin deficiency	
Spleen	Increase: TGF-β1	Sleep fragmentation	
Decrease: IL-1β (SF ob mouse)	Interaction	
EWAT	Increase: IL-1β; TNF-α (SF, wild-type)	Interaction	
Heart	Decrease: IL-1β and TNF-α	Leptin deficiency	
Increase: TGF-β1	Leptin deficiency	
Hippocampus	Increase: IL-1β and TNF-α	Leptin deficiency	
Hypothalamus	Increase: IL-1β and TNF-α	Leptin deficiency	
Prefrontal cortex	Increase: IL-1β and TNF-α	Leptin deficiency	

There were few tissues affected by acute SF in this study, although there were certainly some notable nonsignificant trends that may require further examination (e.g., TNF-α in liver). There was a significant increase in IL-1β expression in epipidymal white adipose tissue among wild-type mice exposed to 24 h of SF (relative to nonsleep-fragmented controls) which is consistent with a previous study in our lab (Dumaine & Ashley, 2015). Among wild-type mice, acute SF also increased TNF-α expression in EWAT as well as increased expression of TGF-β1 in spleen. Interestingly, ob mice were unresponsive to the effects of acute SF compared with wild-type mice. Although further study is needed, perhaps ob mice have pre-existing “metainflammation” (Lumeng & Saltiel, 2011) that precludes further responses from SF (He et al., 2015; Pak, Grandner & Pack, 2014). It is well known that chronic exposure to SF and the hypoxic intervals associated with OSA results in cell damage (Pak, Grandner & Pack, 2014). More specifically, damage from oxidative stress and inflammation in pancreas, liver, heart, and adipose tissue are hypothesized to result in reduced insulin sensitivity, reduced glucose clearance, and disruption of elastic fibers in cardiac vessel endothelium that is associated with chronic SF (Carreras et al., 2014; Jia et al., 2014; Youn et al., 2014; Zhang et al., 2014). Previous literature indicates pro-inflammatory and anti-inflammatory cytokines have increased expression in the tissue of OB individuals compared with lean individuals (Fjeldborg et al., 2014).

Additional studies have shown elevated IL-1β expression in the mouse brain following sleep deprivation, while the expression of TNF-α has varied depending upon the study (Chennaoui et al., 2011; Weil et al., 2009; Zielinski et al., 2014). In contrast to previous literature, leptin deficiency was reported as the only factor that affected pro-inflammatory cytokine gene expression in the brain. Sleep treatment did not significantly affect brain IL-1β or TNF-α gene expression in this experiment. This is surprising considering that previous research has indicated that the same frequency of SF induces neuroinflammation in the hypothalamus (Dumaine & Ashley, 2015).

Chronic SF increases orexigenic behavior, while caloric expenditure remains unchanged, ultimately leading to weight gain from excess food intake (Carreras et al., 2014; Wang et al., 2014). Weight gain is typically mediated by alterations in the expression of leptin and ghrelin, which are satiety and orexigenic hormones, respectively, that act antagonistically on the hypothalamus to maintain energy balance (Garcia-Garcia et al., 2014). Although ob/ob mice are leptin gene KOs, sleep fragmented leptin-deficient mice and wild-type mice lost significant body mass when compared with control mice. Consequently, the alterations in orexigenic behavior following sleep curtailment are likely multifaceted and cannot be explained solely by alterations in ghrelin and leptin, at least in the short term (Pak, Grandner & Pack, 2014). Furthermore, body mass fluctuations following acute SF could also be affected by a physiological stress response or decreased food intake, coupled with increased locomotion during the prolonged period of wakefulness. While weight gain is typically observed in chronic murine SF experiments, this weight gain is time-dependent. The progression toward metabolic syndrome, characterized by reduced glucose clearance and insulin sensitivity, does not alter body mass until multiple days of exposure to SF (Carreras et al., 2014; Dumaine & Ashley, 2015). Consequently, the mass loss observed in leptin-deficient and wild-type mice following acute SF is likely due to a combination of increased activity and reduced food intake. We speculate that the slight mass loss observed in nonsleep-fragmented mice is probably a result of subjects being exposed to a novel environment, which may have produced some anxiety/fear that interrupted normal food intake.

The observed alterations in cytokine gene expression in this experiment were detected at the mRNA level. While previous experimentation has shown that mRNA and protein expression levels of TNF-α are correlated following experimental sleep loss and vaccine stimulation, experiments involving IL-1β have produced conflicting results and TGF-β1 has not been studied within this context (Irwin et al., 2006; Shebl et al., 2010). However, more recent literature suggests that fold changes in protein expression are largely regulated by alterations in mRNA abundance; moreover, other processes, such as protein degradation, may also affect the proteome (Jovanovic et al., 2015). Consequently, it is equivocal whether protein levels correlate directly with mRNA expression, but nonetheless, increased mRNA expression still provides insight into the rapid temporal dynamics of cytokine activation at the transcriptional level.

Sleep treatment altered serum corticosterone concentration independently of initial body mass. Both the wild-type and leptin-deficient SF groups exhibited increased serum glucocorticoid concentrations compared with non-SF controls, but there was no difference detected between the SF groups. SF was the only factor affecting alterations in stress hormone levels, as reported in previous experiments (Ashley et al., 2013; Bonnavion et al., 2015; Nieto et al., 2000). Although ob/ob mice exhibit hypercorticosteronemia, serum corticosterone concentration was not different between OB leptin-deficient and wild-type mice. Furthermore, the increase in serum corticosterone concentration and mass loss as a result of SF is consistent with a previous study (Dumaine & Ashley, 2015). Alternatively, increases in serum corticosterone concentration could have been attributed to two possibilities: (a) a shift in natural biological rhythms from forced arousals during the light phase of the LD cycle and/or (b) an increase in activity levels in the SF groups as a result of tactile stimulation from the swipe bar. The combination of body mass loss and elevated serum corticosterone concentrations suggests that SF mice were experiencing a physiological stress response, including activation of the HPA axis, as a result of sleep interruptions occurring so frequently. A recent study from our laboratory has also demonstrated that exposure to novel environments (such as new cages) can also affect inflammatory and adrenocortical responses to sleep loss (Ashley et al., 2016). Therefore, it is unresolved whether HPA activation occurred because of the method of SF used or the direct effect of disrupted sleep per se.

Conclusion

This experiment provides evidence for differential cytokine gene expression between OB leptin-deficient and wild type mice exposed to acute SF. Since the genetic KO used in this experiment may affect pro-inflammatory cytokine gene expression, replication within the context of a diet-induced model of obesity would be helpful. Presently, the results suggest that a pro-inflammatory environment was induced in the brain of leptin-deficient mice, while a largely anti-inflammatory effect was produced in the periphery, despite heightened serum corticosterone concentrations. Because the response to SF is different between wild-type and leptin-deficient animals, these results may imply different treatment options in terms of therapeutic methods for individuals with OSA. For example, to treat inflammation in OB patients due to leptin resistance, preventative measures, such as reducing body weight through diet and exercise, would likely be effective, rather than solely treating inflammation created as a result of SF. Consistent with our previous work, a single 24 h SF period was sufficient to induce a pro-inflammatory environment in adipose tissue. Since acute sleep loss can lead to chronic effects, manipulating sleep over a longer timeframe will provide needed insights into inflammation-induced pathology associated with chronic conditions, such as OSA.

Supplemental Information

Supplemental Information 1 Raw data for TNF-alpha, heart.

Click here for additional data file.

Supplemental Information 2 Raw data for TGFbeta in heart.

Click here for additional data file.

Supplemental Information 3 Raw data for IL-1beta in heart.

Click here for additional data file.

Supplemental Information 4 Raw data for IL1-beta in peripheral tissues.

Click here for additional data file.

Supplemental Information 5 Raw data for IL1-beta in brain.

Click here for additional data file.

Supplemental Information 6 Raw data for corticosterone levels.

Click here for additional data file.

Supplemental Information 7 Raw data for TGFbeta in brain.

Click here for additional data file.

Supplemental Information 8 Raw body mass data.

Click here for additional data file.

Supplemental Information 9 Raw data for TNF-alpha in brain.

Click here for additional data file.

Supplemental Information 10 Raw data for TNF-alpha in peripheral tissues.

Click here for additional data file.

Supplemental Information 11 Raw data for TFG-beta in peripheral tissues.

Click here for additional data file.

Thanks to Naomi Rowland for assistance with RT-PCR and other protocols and the WKU Biotechnology Center for access to resources.

Additional Information and Declarations

Competing Interests

Author Contributions

Animal Ethics

Data Availability

The authors declare that they have no competing interests.

Jennifer E. Dumaine conceived and designed the experiments, performed the experiments, analyzed the data, prepared figures and/or tables, authored or reviewed drafts of the paper.

Noah T. Ashley conceived and designed the experiments, prepared figures and/or tables, authored or reviewed drafts of the paper.

The following information was supplied relating to ethical approvals (i.e., approving body and any reference numbers):

The Institutional Animal Care Committee at Western Kentucky University provided full approval for this research.

The following information was supplied regarding data availability:

All raw RTPCR, body mass, and ELISA data have been provided as Supplemental Dataset Files.

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
