# Peer review of "Acute sleep fragmentation does not alter pro-inflammatory cytokine gene expression in brain or peripheral tissues of leptin-deficient mice"

_PeerJ, doi:10.7717/peerj.4423_

## Round 0.1 · original submission · Major Revisions

The manuscript is of interest, however all the points raised by the reviewers need careful attention by the authors. The manuscript has to be improved and hopefully the authors will be prepared to perform the required changes an/or additions

Reviewer 1 ·

Basic reporting

In this manuscript ' Sleep fragmentation does not alter peripheral or brain cytokine gene expression of leptin-deficient mice', the authors investigated two factors, acute sleep deprivation and leptin deficiency (obesity) on the production of pro- and anti-inflammatory cytokines in peripheral tissues and the brain. The study shows no significant interaction between sleep fragmentation and leptin deficiency in regulating the inflammatory conditions. Although the results of this study do not conform with some of the published work and did not demonstrate their own hypothesis, it is described with professional langue with sufficient background information, literature research as well as detailed result discussion. The authors further discussed some of the potential caveats at the end of the discussion, which is good for the authors to draw their own conclusions.

Experimental design

One issue with the experimental design is that the authors did not discuss why they only choose acute 24hr sleep deprivation for their study. The longer duration or chronic sleep deprivation monitor may be necessary to draw a more valid conclusion.

Validity of the findings

1. In figure 1, the authors need to explain why without SF treatment, the control group mice also show weight loss.
2. In all figures, the way the authors describe the statistical difference between groups is very confusing. Instead of labeling with the bar with a/b or A/B, it is better with the more common way, asterisks system with lines to indicate between the different groups. In addition, in Figure 2C it is better to move the labels to 2A instead.
3. The authors concluded that leptin deficiency increased both pro- and anti-inflammatory cytokines in all the test areas of the brain. This is interesting but could not be easily explained, as of how can one factor leads to complete opposite effects at the same time. The authors need to explain this confounding results or further validate their data.

Additional comments

I would like to thank the editor for inviting me to conduct the review work. I really appreciate this chance to work with you and am looking forward to more opportunities to contribute to PeerJ.

Reviewer 2 ·

Basic reporting

In this study, the authors used leptin-deficient mice to study the relationship between sleep fragmentation and cytokine gene expression. Overall, the authors provided sufficient discussion in the text. However, I have some concerns about this work.
1. In general, the whole text is fluent. But I think the author could do a better job if they revise and improve the language more carefully. There are still some grammatical mistakes in the paper. Meanwhile, some sentences are difficult to understand. My suggestion is to use simple sentences if the authors have any difficulties in writing complex and long sentences.
2. The authors should improve the introduction to tell the readers about pro-inflammatory cytokines, especially the genes that they studied in this work. More importantly, since they authors only studied 3 genes and several tissues in this work, is that appropriate to claim "does not alter peripheral or brain cytokine gene expression" in the title?
3. One of my major concerns is about the sleep status in leptin-deficient mice. As the authors mentioned in the text, previous study has shown that leptin plays an important role in regulating the sleep status and the leptin-deficient mice exhibited abnormal sleep cycle. I wonder whether it is appropriate to apply artificial sleep fragmentation on those animals since they already have problems in sleep. The authors should clarify and discuss this point.
4. It seems that in many samples (figure 2 and figure 3), the OB SF has a trend to increase or decrease. The huge variation in each group may eliminate the statistic difference. I think it is reasonable to add more animals into each group to make the conclusion more convincing.
5. Figure 1. The sample size of OB SF is 9 in the raw data. But it is 8 in the figure 1. The authors should clarify this.
6. The authors should mention the sample size of Figure 2 in the text.
7. Since the qPCR is the key experiment in this study, the authors should provide the data of primer validation, such as the images of a single band of the PCR product on the gel.

Experimental design

no comments

Validity of the findings

no comments

Additional comments

no comments

---

## Round 0.2 · accepted · Accept

The manuscript has been improved following the changes required by the reviewers

Reviewer 2 has made a couple of suggestions which can be addressed while in production

Reviewer 2 ·

Basic reporting

Overall, I am satisfied with the revised version. I only have two minor comments:
1. Line 105-106: "This value exceeds by an order of magnitude the number of baseline arousals that typically occur in ob/ob and wild-type mice, respectively, in similar conditions". The authors should re-write this sentence to make their point clear.
2. Figure 1 and Figure 4: As the authors mentioned in the responses, one mouse in the OB SF group was removed during the experiment due to health reasons. But it seems that they still used that mouse in Figure 4. The author should clarify this point.

Experimental design

no comments

Validity of the findings

no comments

Additional comments

no comments